# Effects of a joint outdoor exercise program for dog owners and dogs on physical activity, sedentary time and sleep-related behaviors

Klara Smedberg[1]*, Anna Bergh[2], Erika Roman[1,3], Sören Spörndly-Nees[4,5], Jan W. Eriksson[6], Lena V. Kallings[7,8], Josefin Söder[2]

1 Department of Animal Biosciences, Swedish University of Agricultural Sciences, Uppsala, Sweden, 2 Department of Clinical Sciences, Swedish University of Agricultural Sciences, Uppsala, Sweden, 3 Department of Pharmaceutical Biosciences, Uppsala University, Uppsala, Sweden, 4 Department of Women´s and Children´s Health, Physiotherapy and Behavioral Medicine, Uppsala University, Uppsala, Sweden, 5 Primary Care and Health, Region Uppsala, Uppsala, Sweden, 6 Department of Medical Sciences, Uppsala University, Uppsala, Sweden, 7 Department of Physical Activity and Health, The Swedish School of Sport and Health Sciences, Stockholm, Sweden, 8 Primary Care, Department of Public Health and Caring Sciences, Uppsala University, Uppsala, Sweden

* klara.smedberg@slu.se

## Abstract

Lack of physical activity (PA) is a rising health challenge in both humans and companion dogs. Therefore, One Health strategies to increase PA in both species have been suggested. The aim of this pilot study was to evaluate the effects of a joint outdoor exercise program with jogging and circuit training sessions for dog-dog owner pairs (n = 15) on PA, sedentary time and sleep-related behaviors. Furthermore, the study aimed to assess fulfilment of global recommendations on PA in dog owners, and to compare accelerometer-derived and self-reported activity levels. Body measurements were used as secondary outcomes. Results showed an increase of vigorous-intensity PA by 5 min per day (P = 0.04) and a reduction of sedentary time by 41 min in dog owners (P = 0.01). Accelerometer data showed a high degree of fulfilment of recommendations for physical activity in dog owners at baseline. Self-reported PA and sedentary time were underestimated compared to accelerometer-derived data. No major changes in activity patterns in dogs were detected, but there was a slight but significant reduction of median body condition score. Sleep-related data indicated satisfactory nighttime sleep patterns in both species, with no considerable effects of the program. Both dog owners and dogs spent the main part of the daytime period sedentary. The exercise program may be useful for increasing vigorous PA and reducing sedentary time in dog owners. The high levels of PA in dog owners are in line with previous findings, and the underestimation of self-reported moderate-intensity PA suggests a strong integration of dog walking into daily routines. However, the high levels of sedentary time underline the importance of assessing sedentary behavior in addition to PA in both species in future studies. The

**Data availability statement:** The authors confirm that the data supporting the findings of this study are available within the article and its supplementary material. To adhere to the ethical permit and not compromise the privacy of participating dog owners, dog owner raw data is provided in de-identified form.

**Funding:** The salary of K.S. was funded by the Uppsala Diabetes Centre (UDC), Sweden (https://www.uu.se/centrum/diabetes). J.S. received research grants from the Research Foundation of Agria and the Swedish Kennel Club (https://www.skk.se/agria-och-skk-forskningsfond/) and A.B. received research grants from SLU Future One Health, Swedish University of Agricultural Sciences (https://www.slu.se/en/about-slu/organisation/future-platforms/slu-future-one-health/). J.W.E. and E.R. had funding from the Swedish Diabetes Foundation (https://www.diabetes.se/forskning/diabetesfonden/, award number DIA2021-661, DIA2024-935), the Swedish Research Council (https://www.vr.se/english.html, award number 2024-03344) and the Swedish Foundation for Strategic Research (https://strategiska.se/en/, award number CMP22-0014). J.W.E. also had funding from the Novo Nordisk Foundation (https://novonordiskfonden.dk/en/, award number NNF23OC0084483, NNF25OC0101843) and the European Commission Horizon RIA project PAS GRAS (https://cordis.europa.eu/project/id/101080329, award number 101080329). The funders had no role in study design, data collection and analysis, decision to publish, or preparation of the manuscript.

**Competing interests:** The authors have declared that no competing interests exist. The funders had no role in study design, data collection and analysis, decision to publish, or preparation of the manuscript.

findings should be confirmed in randomized controlled studies with larger sample size and long-term follow-up.

## Introduction

Increasing levels of physical inactivity and sedentary behavior is a major health concern in humans. Physical inactivity increases the risk for non-communicable diseases such as obesity, cardiovascular disease, type 2 diabetes, dementia and several cancer forms [1]. Furthermore, high levels of sedentary time are associated with increased all-cause mortality [2]. Physical inactivity is also detrimental for quality of life and mental health [3]. Moreover, sound habits of physical activity (PA) are of importance for sleep outcomes, as there is strong evidence that both regular and acute bouts of PA improve sleep parameters in humans [4]. Even though people who own a dog generally have higher levels of PA [5], it has been shown that more inactive dog owners have inactive dogs [6]. A lack of PA in companion dogs has been associated with increased prevalence of canine overweight and obesity [7,8]. As in humans, overweight in dogs is linked to metabolic disturbances and chronic diseases [9–11], and canine obesity is associated with a reduced quality of life [12] and a shortened lifespan [13].

The World Health Organization (WHO) recommends that adults aged 18–64 should perform at least 150–300 min of moderate-intensity PA, 75–150 min of vigorous-intensity PA, or an equivalent combination throughout the week [14]. The global prevalence of insufficient PA, defined as the proportion of the population not meeting these recommendations, was estimated to 31.3% in 2022, which is an increase from previous estimates of 23.4% from 2000 and 26.4% from 2010 [15]. To increase PA at all levels in society, the WHO has adopted the Global Action Plan on Physical Activity 2018–2030 [16], and encourages all countries to tackle the burden of physical inactivity with innovative solutions [17]. One such innovative solution may be One Health approaches that encompass both humans and their companion animals. A substantial proportion of households in the Western world have a companion dog [18–20], but it has been estimated that up to half of dog owners do not walk with their dogs regularly [21]. In addition, it has been shown that dog owners' exercise routines influence the equivalent in their dogs, and that owners who perform vigorous exercise are more likely to have dogs that also perform vigorous exercise [6]. These findings highlight the value of One Health strategies to mutually increase PA in both species. Thus, dog owners constitute a considerable target group for strategies to increase PA. Accelerometry is widely used for more objective assessment of PA, sedentary time and sleep-related behaviors in humans [22–25]. In addition, it has been shown to be a valid and reliable tool for assessing PA and sedentary time in dogs [26], and has in recent years increasingly been used for assessing canine sleep-related behaviors [27,28]. The aim of this pilot study was therefore to evaluate the effects of a joint outdoor exercise program for dog-dog owner pairs on PA, sedentary time and sleep-related behaviors in both species, while using body measurements as secondary outcomes. Furthermore, the study aimed to assess dog owners'

fulfilment of WHO recommendations on PA, and to compare accelerometer-derived and self-reported levels of PA and sedentary time.

## Methods

### Ethical statement

This study was approved by the Ethics Committee for Animal Experiments, Uppsala, Sweden (Dnr 5.8.18–15533/2018) and by the Swedish Ethical Review Authority, Stockholm, Sweden (Dnr 2021−01014). It adhered to the Declaration of Helsinki, the guidelines of the Swedish Legislation on Animal Experimentation (Animal Welfare Act SFS 2018:1192) and the European Union Directive on the Protection of Animals Used for Scientific Purposes (Directive 2010/63/EU). All dog owners provided informed written consent prior to data collection.

### Study population

**Dog owners.** Dog owners were recruited on a non-randomized basis through calls in social media and on the websites of the Swedish University of Agricultural Sciences (SLU) and the Swedish Working Dog Association. Inclusion criteria were age ≥ 18 years and in physical and mental condition allowing participation at the lowest level in the exercise program (2 km jogging). The exclusion criteria were self-reported ischemic heart disease or heart failure, active malignancy, type 1 diabetes, chronic pulmonary diseases or severe psychiatric disorders, including alcohol and substance use disorders.

**Dogs.** Inclusion criteria for dogs were age ≥ 1 year and in physical condition allowing participation at the lowest level in the exercise program. Exclusion criteria included known systemic or orthopedic diseases that could pose a health risk when participating in the study or known aggressiveness or timidity that could affect the ability to be handled by researchers. All dogs participated in the study with their owner or handler and had to pass a clinical veterinary examination with specific focus on orthopedic status before the start of the exercise program.

### General study protocol

The overall experimental outline is shown in Fig 1. In short, dog owners and dogs participated in an eight-week joint outdoor exercise program designed by the Swedish Working Dog Association, with the overarching aim of promoting outdoor PA, health and wellbeing in dog owners and dogs [29]. Data on PA, sedentary time and sleep-related behaviors (henceforth collectively referred to as "PA behaviors") in dog owners and dogs were collected from tri-axial accelerometers. Accelerometers were worn on 24-hour basis during two time periods: the week before the start of the exercise program (henceforth referred to as baseline) and the eighth and final week of the exercise program (henceforth referred to as intervention period). During these periods, dog owners were also asked to keep activity and sleep diaries, and to provide questionnaire-based data on self-assessed PA, sedentary time, sleep quality and sleep routines, as well as perceived physical capacity in themselves and their dogs. Body measurements (body mass index and waist-hip ratio in dog owners and body condition score in dogs) were used as secondary outcomes. In addition, dogs underwent clinical examinations at both registration periods. The study was conducted in Uppsala, Sweden during August to October 2021.

**Joint outdoor exercise program.** The eight-week outdoor exercise program included joint jogging and circuit training sessions for dog owners and dogs, as previously described by Smedberg et al. (2024) [30]. Dog owners individually selected a suitable target jogging distance of 2, 5, 7.5 or 10 km, with respect to previous exercise experience and physical capacity in both themselves and their dogs, and with the goal of being able to jog the selected distance with the dog at the end of the intervention period. Jogging sessions were performed twice a week, except for participants aiming for 10 km, who performed three sessions per week. Distance and intensity were gradually increased throughout the exercise program. Participants aiming for distances of 2, 5 and 7.5 km were instructed to progressively increase intensity by alternating jogging and walking until the final week of the program, while participants aiming for 10 km increased intensity

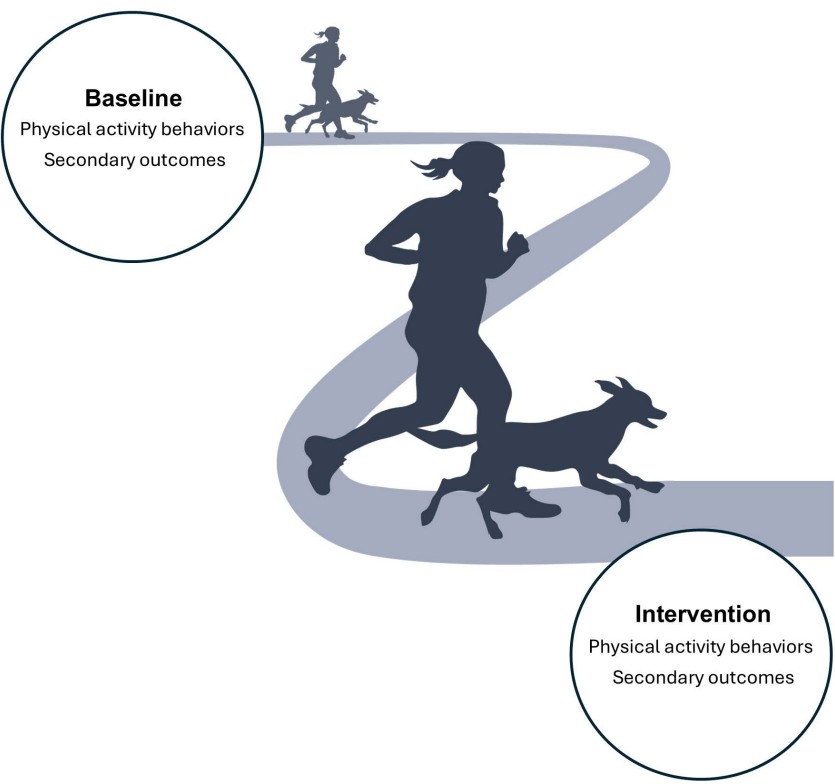

**Fig 1. Experimental outline of the joint outdoor exercise intervention for dog owners and dogs.** Accelerometer-derived and self-reported physical activity behaviors (physical activity, sedentary time and sleep-related behaviors) were used as outcome measures during baseline and intervention periods, along with secondary outcomes (body mass index and waist-hip ratio in dog owners and body condition score in dogs).

through interval sessions with gradually decreasing resting periods. Circuit training exercises were conducted once a week for all participating dog owners and dogs. Six different training exercises were performed at each session, four strength exercises focusing on leg/hindleg, arms/foreleg, core and neck muscles and two exercises focusing on agility and explosive speed. Time for each exercise gradually increased from 30 to 60 seconds during the program.

## Outcome measures

**Accelerometer model and placement in dog owners and dogs.** Tri-axial accelerometers (ActiGraph GT3X, ActiGraph LCC, Pensacola, FL, USA) were used to measure PA behaviors. Dog owners were instructed to wear the accelerometer on a 24-hour basis during baseline and intervention periods (seven days per period). The accelerometer was to be worn on the right side of the waist during the day and on the non-dominant wrist during the night. Dog owners were instructed to equip their dog with a provided neck collar with a pre-attached accelerometer during the same two time periods. The collar was to be worn with the accelerometer placed on the underside of the neck of the dog. For both parties, the accelerometer was to be removed during water-based activities, such as showering or swimming. If the accelerometer was removed for any reason, dog owners were instructed to note this in the activity diaries.

**Accelerometer settings, data extraction and data validation procedures in dog owners and dogs.** ActiLife (version 6.13.4, ActiGraph LCC, Pensacola, FL, USA) software was used for initializing the accelerometers and for extracting and processing the collected data. Data was exported to Microsoft Excel (Microsoft 365, Redmond, WA, USA) for further handling. Accelerometers were set at 100 Hz. Tri-axial raw data were combined to a single resultant vector,

extracted in 10-second epochs using the software's normal frequency filter and expressed as counts per min (cpm). Patterns of PA and sedentary and sleep-related behaviors are presented as average time spent in predefined intensity categories per day, as recommended by Arvidsson et al. (2024) [31]. In addition, PA intensity is presented as vector magnitude cpm per hour. The daily mean for each participant was used to calculate cohort-level results.

For validation of dog owner data, the Troiano (2007) wear time validation parameters in ActiLife were used, defining non-wear time as 60 consecutive min with no movement (0 cpm), with an allowance for up to two min of intensities up to 100 cpm. As there are no established canine wear time validation parameters, non-wear time in dogs was defined as 60 consecutive min with no movement (0 cpm), with no allowance for spikes of activity. Non-wear hours from both dog owners and dogs were removed from the data before further analyses were performed. Based on previous studies, participants with a minimum of 600 min (10 h) of valid daytime wear time for at least four days (dog owners) [32] or three days (dogs) [33] during both registration periods were included in the analysis.

**Accelerometer data handling in dog owners.** Patterns of PA and sedentary behavior in dog owners were assessed during awake hours only. Nighttime sleep periods were detected and removed according to the following procedure: sleep periods as detected by ActiLife were checked against the owners' sleep diary notations. If there was a discrepancy between these two measures, the sleep diary notations were used to define the night in question. Nighttime sleep periods were then manually removed. Physical activity behaviors during awake hours were divided into the following intensity categories: sedentary behavior, light-intensity PA (LPA), moderate-intensity PA (MPA), vigorous-intensity PA (VPA) and moderate- to vigorous-intensity PA (MVPA, defined as the sum of MPA and VPA) [34,35]. Definitions and cut-offs for the intensity categories are shown in Table 1. Prolonged sedentary bouts were defined as ≥ 20 min below 200 cpm [36]. For analysis of nighttime sleep-related behaviors, the ActiLife Tudor-Locke auto sleep period detection function, based on the Cole-Kripke sleep algorithm [25,37], was used. As the ActiLife software requires 60-second epoch files to run the sleep period detection function, accelerometer raw data was reintegrated to 60-second epochs for the purpose of sleep analysis. The following sleep parameters were calculated: sleep efficiency, total minutes in bed, total sleep time, wake after sleep onset, number of awakenings, average awakening length, fragmentation index, movement index and sleep fragmentation index.

Definitions and cut offs for physical activity behaviors in dog owners and dogs are based on WHO guidelines on physical activity and sedentary behavior (2020) [14], Sasaki et al. (2011) [34], Aguilar-Farias et al. (2014) [35], Morrison et al. (2013) [7], Ladha and Hoffman (2018) [27] and Hoffman et al. (2020) [28]. Sedentary behavior in dogs is presented both in total and divided into the subcategories "sleep-related sedentary behavior" and "non-sleep-related sedentary behavior". Abbreviations: cpm, counts per minute; LIPA, light-intensity physical activity; LMPA, light- to moderate-intensity physical activity; MPA, moderate-intensity physical activity; MVPA, moderate- to vigorous-intensity physical activity; VPA, vigorous-intensity physical activity.

**Accelerometer data handling in dogs.** Due to the lack of validated canine sleep algorithms, dog data were separated into predefined day- and nighttime periods to make distinctions of canine patterns of PA and sedentary behavior in daytime versus nighttime. Daytime for dogs was defined as 07.00 a.m. to 22.59 p.m. and nighttime as 23.00 p.m. to 06.59 a.m. The following intensity categories were used for PA behaviors in dogs: sedentary behavior, light- to moderate-intensity PA (LMPA) and VPA [7]. Inspired by findings by Ladha and Hoffman (2018) [27] and Hoffman et al. (2020) [28], sedentary behavior is presented both in total and divided into the subcategories sleep-related sedentary behavior and non-sleep-related sedentary behavior. Definitions and cut-offs for the intensity categories are shown in Table 1. Sleep-like bouts in dogs were defined as ≥ 10 min in sleep-related sedentary behavior.

**Data handling regarding dog owner fulfilment of WHO recommendations for PA.** Accelerometer-assessed dog owner time in MPA, VPA and MVPA was categorized according to WHO recommendations for PA, which state that adults ≥ 18 years should perform at least 150–300 min of MPA, or at least 75–150 min of VPA, or an equivalent combination of MPA and VPA (in this study equated to MVPA) per week [14]. Since the dog owners had less than seven

**Table 1. Intensity categories for physical activity behaviors in dog owners and dogs.**

| Intensity categories | Definitions | Cut-off (cpm) |
|---|---|---|
| Dog owners | | |
| Sedentary behavior | Activities such as sitting during office-based work, watching television or driving a car, reclining, or lying during awake hours | ≤ 199 |
| LPA | Activities that do not result in a substantial increase in heart rate or breathing rate | 200–2689 |
| MPA | Activities representing a 5 or 6 on a scale of 0–10 relative to an individual's personal capacity | 2690–6166 |
| VPA | Activities representing a 7 or 8 on a scale of 0–10 relative to an individual's personal capacity | ≥ 6167 |
| MVPA | The sum of MPA and VPA | ≥ 2690 |
| Dogs | | |
| Sedentary behavior | No movement of the trunk, including during sleep | ≤ 1351 |
| Sleep-related sedentary behavior | Only movements so subtle that they are difficult to detect for the human eye, indicating sleep or a sleep-like state | ≤ 150 |
| Non-sleep-related sedentary behavior | Any sedentary behavior exceeding that defined as sleep-related sedentary behavior | 151–1351 |
| LMPA | Slow to moderate translocation of the trunk, while confined within a room or kennel, or outdoors with the dog on leash | 1352–5695 |
| VPA | Rapid translocation of the trunk while running outdoors and off leash | ≥ 5696 |

days of valid data per registration period, their daily average of minutes spent in the specified intensity categories was multiplied by seven to assess the proportion meeting six different criteria: 150 (lowest achievable criterion) and 300 (highest achievable criterion) min of MPA per week, 75 (lowest achievable criterion) and 150 min (highest achievable criterion) of VPA per week, as well as 150 and 300 min of MVPA per week. The WHO presents no specific limit for daily sedentary time in their guidelines. Based on Ekelund et al. (2019) [2], a cut-off of sedentary time of ≥ 9.5 h was used to define high sedentary time in this study. For assessment of the number of dog owners exceeding the 9.5 h cut-off, only data from awake hours were used.

**Diaries.** Dog owners were asked to keep activity and sleep diaries on the same days that the accelerometers were worn. Diaries were provided in paper form, and data was manually transferred to Microsoft Excel by the first author after study completion.

In the activity diaries, dog owners were instructed to note date, type of activity, duration in minutes, self-assessed intensity (light, moderate or vigorous) for all types of physical activities performed, including household work, gardening and other non-sport activities. Levels of diary-derived PA were used for comparisons with accelerometer-derived PA data, and for this reason, diary notes for days without valid accelerometer data were excluded from analysis. Hence, comparisons of diary- and accelerometer-derived PA were based on the exact matched days. Since dog owners were not asked to note sedentary time in the diaries, comparisons were not made for sedentary behavior.

In the sleep diaries, dog owners were instructed to note what time they went to and out of bed, an estimation of the time it took for them to fall asleep, as well as the number and length of awakenings (if any) during the night. Sleep diary data was used for confirmation of accelerometry-detected sleep periods in dog owners.

**Questionnaires.** Self-reported data on PA and sedentary time and sleep-related behaviors in dog owners, as well as perceived physical capacity in dog owners and dogs, were collected during both registration periods. All questionnaires

were designed in the digital software Netigate (Netigate AB, Stockholm, Sweden), and the collected data was exported to Microsoft Excel for further handling. All questionnaires are included in S1 File.

Two questions were used for self-assessments of weekly PA time and one for daily sedentary time in dog owners, as previously described by Smedberg et al. (2024). In short, the PA questions were designed by the Swedish National Board of Health and Welfare and distinguish between time spent on everyday PA, such as walking or gardening, and time spent exercising [38]. The latter is defined as PA resulting in shortness of breath, such as jogging and other sport activities. For assessment of daily sedentary time (sleep excluded), the sitting single-item question developed by Kallings et al. (2019) [39] was used. Levels of self-assessed questionnaire-derived PA and sedentary time were used for comparisons with accelerometer-derived data. Everyday PA was equated with MPA as defined by WHO, and exercise was equated with VPA [14]. As time spent in PA was estimated per week in the questionnaires, a daily average was calculated for comparisons with daily accelerometer-derived data.

For self-assessment of perceived physical capacity in dog owners, the rating of perceived capacity (RPC) scale was used [40]. Dog owners were asked to rate their physical capacity by assessing on a scale from 1 to 20 which activity (ranging from sitting to performing aerobic exercise at professional level) they were able to perform for at least half an hour. They were also asked to make the corresponding assessment for their dogs on a modified rating of perceived capacity scale from 1 to 5 (created for the purpose of this study) with activities ranging from sitting to running at high speed.

For self-assessment of sleep-related behaviors in dog owners, three validated questionnaires were used: the Insomnia Severity Index (ISI) [41], the Epworth Sleepiness Scale (ESS) [42] and the Pittsburgh Sleep Quality Index (PSQI) [43]. In addition, three questions regarding co-sleeping with the dog, created for the purpose of this study, were used: dog owners were asked to what extent they shared bedroom or bed with their dog, and how often they were disturbed at night by their dog.

**Secondary outcomes and clinical examination.** Body mass index (BMI) and waist-hip ratio (WHR) in dog owners and body condition score (BCS) in dogs were used as secondary outcomes. Body mass index (BMI) was calculated as described by the WHO by dividing body weight (kg) by height (m) square [44]. Height and bodyweight were measured to the nearest mm and hg, respectively. Waist and hip circumferences were measured in triplicates to the nearest 0.5 cm. Waist-hip ratio (WHR) was calculated as described by the WHO by dividing mean waist circumference by mean hip circumference [45].

All BCS assessments were performed by the same veterinarian with specific expertise. Assessments were conducted according to guidelines from the World Small Animal Veterinary Association (WSAVA) based on the 9-point BCS scale developed by Laflamme (1997) [46], in which 1–3 represent underweight, 4–5 ideal weight, 6 slight overweight, 7 overweight and 8–9 represent obesity. The dogs underwent clinical examinations before and after the exercise program, by the same veterinarian and with specific focus on orthopedic status.

**Data processing and statistical analyses.** This study is based on quantitative data from accelerometry, diary notes, questionnaires and body measurements. Microsoft Excel and GraphPad Prism (version 10.0, San Diego, CA, USA) were used for data processing and statistical analyses. GraphPad Prism, Microsoft PowerPoint (Microsoft 356, Redmond, WA, USA) and Adobe Illustrator (version 29.8.1, San José, CA, USA) were used for creation of figures. Shapiro-Wilks test was used for testing of normal distribution of data. Wear time data in minutes per day was normally distributed for dog owners at both registration periods, but not for dogs. The following dog owner data did not follow normal distribution at both registration periods: accelerometer-derived data regarding PA intensity and the proportion spent in sedentary bouts, total sleep time, sleep fragmentation and movement index, questionnaire-derived data regarding MPA, VPA and sedentary time, diary-derived LPA, MPA and VPA data, RPC scale data, BMI data, ISI data and PSQI data. In dogs, accelerometer-derived data regarding PA intensity, sedentary time, LMPA and VPA and the proportion of nighttime hours spent in sedentary behavior did not follow normal distribution, neither did BCS and modified RPC scala data. Analyses in which

not all related parameters were normally distributed at both registration periods were performed with Wilcoxon matched-pairs signed rank test, while one sample T-tests were used when all related parameters were normally distributed at both periods. Paired analyses were two-tailed. Results are presented as mean value ± standard deviation (SD) for normally distributed data and median, minimum and maximum values for not normally distributed data. The threshold for statistical significance was set to $P < 0.05$ in all analyses. Descriptive statistics were used for calculations of dog owner fulfilment of recommendations on PA and sedentary time, and for data regarding habits of co-sleeping with the dog.

## Results

### Study population

The data collection resulted in valid accelerometer registrations and complete questionnaires from 15 dog-dog owner pairs; see Fig 2 for recruitment and drop-out details. The reasons for drop out after completing baseline questionnaires were unknown. Ten pairs in total dropped out during the exercise program; two pairs due to temporary injuries (in one dog and one dog owner, respectively) not related to the intervention, one pair due to a tendon injury in the dog owner, and seven pairs for unknown reasons. An additional four pairs were excluded as they did not fulfil the criteria for minimum accelerometer wear. Complete diary notes were provided by 13 of the 15 dog owners that finished the exercise program with valid accelerometer registrations, and as such, comparisons between accelerometer- and diary-derived data are based on data from a sub-cohort of 13 dog owners.

Baseline demography of participating dog owners (n = 15) and dogs (n = 15) is presented in Table 2. The mean age of dog owners was 47 ± 12 years (mean ± SD), and of dogs 4 ± 2 years. Dogs of 14 different breeds participated in the study. Five dog owners selected the target jogging distance 2 km, six selected 5 km, three selected 7.5 km and one selected 10 km. Two dog owners registered that they changed target distance during the course of the intervention: one from 7.5 to 5 km and another from 5 to 7.5 km. The participant that changed to a shorter target distance stated that this was due to a minor muscle injury in herself. Three dog owners stated that they had a temporary break during the exercise program due to private reasons, temporary illness/injury or a lack of time. All but two dog owners (both in the 2 km group) stated that they managed to reach their selected target distance at the end of the intervention. Median BMI in dog owners was 26.2 (range 19.1–31.5) at baseline, with no significant change at the intervention period (median 26.1, range 19.4–31.7, $P = 0.16$). At the intervention period, 3/15 dog owners were classified as overweight and 5/15 as obese. Neither were there any significant changes in WHR (mean ± SD 0.9 ± 0.1 at both registration periods, $P = 0.80$). In dogs, median BCS at baseline was 5 (range 4–7), which was significantly reduced to a median of 5 (range 4–6) at the intervention period ($P = 0.02$). A reduced BCS was registered in 7/15 dogs, and 13/15 dogs were assessed as having an ideal BCS (BCS 4–5) at the intervention period. None of the participating dogs showed deviations on the initial clinical examination that hindered participation in the intervention, nor did the exercise program result in any adverse effects related to the locomotor apparatus.

A BMI range of 18.5 to < 25 is considered normal, ≥ 25 to < 30 as overweight and ≥ 30 as obesity [44]. BCS 1–3 represent underweight, 4–5 ideal weight, 6 slight overweight, 7 overweight and 8–9 obesity [46]. For WHR, cut-off thresholds of at least 0.90 for men and at least 0.85 for women predict a substantially increased risk of metabolic complications [45]. Represented breeds were: Welsh springer spaniel, Lagotto Romagnolo, golden retriever, Icelandic sheepdog, whippet, bullmastiff, hovawart, boxer, Siberian husky, schapendoes, medium poodle, German shepherd, Smaland hound and mixed breed. Abbreviations: BCS, body condition score; BMI, body mass index; WHR, waist-hip ratio

### Physical activity behaviors in dog owners

**Accelerometer-derived patterns of physical activity behaviors in dog owners.** At baseline, dog owners wore the accelerometers for a median of 6 valid days (range 5–6 days), with a mean ± SD awake wear time of 16.0 ± 1.1 h per day. At the intervention period, median wear time was 5 valid days (range 5–6 days), with a mean ± SD awake wear time of 15.0 ± 0.9 h per day. One weekend day per registration period was included for all participating dog owners.

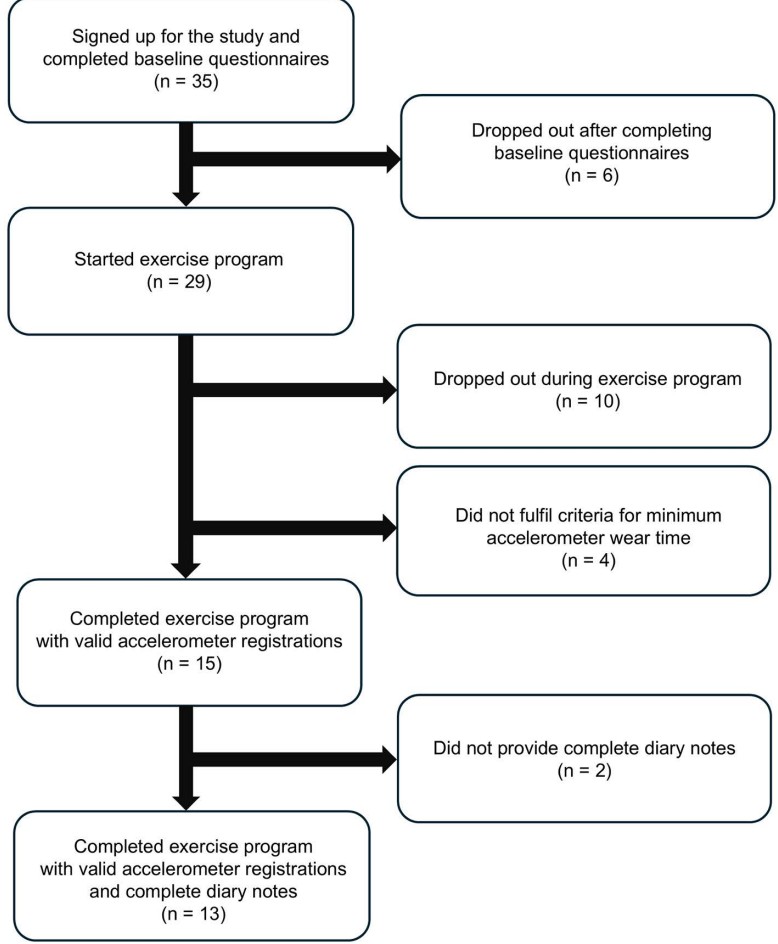

**Fig 2. Flowchart over dog owner-dog pair recruitment and drop-outs.**

Details of patterns of PA behaviors during awake hours in dog owners are presented in Table 3. There was a significant increase in mean VPA of 5 min per day at the intervention period, and a reduction of mean sedentary time of 41 min per day ($P<0.05$). LPA, MPA and MVPA remained unchanged ($P>0.05$). Neither did median PA intensity change; 808 cpm (range 514–1258) at baseline vs 898 cpm (range 463–1083) at the intervention period ($P=0.08$). Dog owners spent 62% of their awake hours sedentary at both registration periods, out of which 15 and 17%, respectively, were spent in prolonged sedentary bouts. Dog owners spent approximately 7.5 h per night in bed and slept just under 7 h per night during both registration periods, resulting in a sleep efficiency of over 90%. There were no significant changes in sleep parameters, except for a slight decrease in average awakening length of 24 sec ($P<0.05$). Sleep parameters for dog owners are presented in S1 Table.

Results from one sample t-test comparing baseline and intervention data. All data were normally distributed, and results are presented as mean ± SD. Significant results are noted in italics. Abbreviations: LPA, light-intensity physical activity; MPA, moderate-intensity physical activity; MVPA, moderate- to vigorous-intensity physical activity; VPA, vigorous-intensity physical activity. Note, mean awake wear time was approximately one hour shorter per day at the intervention period.

**Dog owner fulfilment of WHO recommendations on physical activity.** Based on accelerometer-derived data, all dog owners (n = 15) met the criteria of 150 and 300 min of MPA and MVPA per week, respectively, at both baseline

**Table 2. Baseline demography of participating dog owners (n = 15) and dogs (n = 15).**

| Dog owners | Total number (n) | Proportion (%) |
|---|---|---|
| Gender | | |
| Females | 12 | 80 |
| Males | 3 | 20 |
| Other | 0 | 0 |
| Age | | |
| 18–29 years | 1 | 7 |
| 30–44 years | 3 | 20 |
| 45–64 years | 11 | 73 |
| BMI | | |
| < 18.5 | 0 | 0 |
| 18.5–<25 | 7 | 47 |
| ≥ 25–<30 | 2 | 13 |
| ≥ 30 | 6 | 40 |
| WHR | | |
| < 0.85 (women)<br>< 0.90 (men) | 5 | 33 |
| ≥ 0.85 (women)<br>≥ 0.90 (men) | 10 | 67 |
| **Dogs** | **Total number (n)** | **Proportion (%)** |
| Sex | | |
| Intact females | 4 | 27 |
| Intact males | 5 | 33 |
| Neutered females | 4 | 27 |
| Neutered males | 2 | 13 |
| Age | | |
| 1–3 years | 7 | 47 |
| 4–7 years | 7 | 47 |
| > 8 years | 1 | 7 |
| BCS | | |
| 1–3 | 0 | 0 |
| 4–5 | 10 | 67 |
| 6 | 4 | 27 |
| 7 | 1 | 7 |
| 8–9 | 0 | 0 |

**Table 3. Accelerometer-derived patterns of physical activity behaviors in dog owners (n = 15) during awake hours.**

| Intensity category | Baseline, min per day | Intervention, min per day | P-value |
|---|---|---|---|
| Sedentary behavior | 597 ± 71 | 556 ± 62 | *0.01* |
| LPA | 241 ± 58 | 220 ± 63 | 0.17 |
| MPA | 108 ± 34 | 103 ± 28 | 0.48 |
| VPA | 15 ± 9 | 20 ± 10 | *0.04* |
| MVPA | 122 ± 36 | 123 ± 31 | 0.93 |

and intervention periods. For recommendations of VPA, the number of dog owners fulfilling the criterion of 75 min per week increased from 8 to 14 at the intervention period, and from 3 to 6 for the 150 min criterion. Ten dog owners had a daily sedentary time exceeding 9.5 h at baseline, with a decrease to six dog owners at the intervention period.

**Comparisons of accelerometer-derived and self-assessed patterns of physical activity behaviors in dog owners.** At baseline, comparisons of accelerometer- and questionnaire-derived PA data showed that dog owners (n = 15) significantly underestimated daily time spent in MPA, with a median difference of 60 min ($P < 0.0001$). A corresponding analysis showed no statistically significant difference in VPA; the median underestimation was 26 s ($P = 0.06$). At the intervention registration period, there was a significant underestimation of daily time spent in MPA, with a median difference of 70 min ($P < 0.0001$). Daily time spent in VPA at the intervention period was significantly underestimated, with a median difference of 60 s ($P = 0.01$). There were significant underestimations of the time spent sedentary at both registration periods, with a median difference of 1 h 37 min per day for baseline ($P = 0.04$) and of 4 h 5 min for the intervention registration period ($P = 0.002$).

Comparisons of accelerometer- and diary-derived PA data from a sub-cohort of dog owners (n = 13) showed that they significantly underestimated baseline time spent in LPA by 2 h 40 min ($P = 0.0002$), in MPA by 1 h 30 min ($P = 0.0002$) and in VPA by 13 min ($P = 0.005$). Similar underestimations were registered at the intervention period; by 2 h 48 min for LPA ($P = 0.0002$), by 1 h 25 min for MPA ($P = 0.0005$) and by 11 min for VPA ($P = 0.006$). There were no diary-based data on sedentary time.

**Perceived physical capacity in dog owners.** On the RPC-scale, dog owners rated a physical capacity of 8 in median (range 5–15) at baseline, representing being able to jog or cycle for at least half an hour. At the intervention period, the corresponding median was 9 (range 7–15), which was a non-significant change ($P = 0.09$).

**Questionnaire-derived data on sleep-related behaviors in dog owners.** The results from the three validated sleep questionnaires are presented in S2 Table. Low levels of problems related to insomnia, sleep quality or sleepiness were registered on a cohort level, with no significant changes at the intervention period (P > 0.05). Regarding habits of co-sleeping with the dog, the vast majority of dog owners (12/15 at baseline, 14/15 at intervention) had their dog in their bedroom at least three nights a week, while somewhat fewer also had the dog in their bed (9/15 at baseline, 6/15 at intervention). Most dog owners stated that they never or rarely woke up during the night because of the dog (13/15 at both registration periods).

## Physical activity behaviors in dogs

**Accelerometer-derived patterns of physical activity behaviors in dogs.** Dog median wear time was 5 valid days for both baseline (range 4–6 days) and intervention periods (range 4–5 days). Median wear time per 24-hour period for dogs (nights included) was 24 h (range 22–24 h) at baseline and 23 h 36 min (range 22 h 5 min–24 h) at the intervention period. One weekend day per registration period was included for all participating dogs, except for one dog that did not provide any weekend data for the baseline period.

Details on PA behaviors during the defined day and night in dogs are presented in Table 4. During the defined day, dogs had a median of approximately 3 h of LMPA per day, and a median of 24 min of VPA, with no significant changes between registration periods (P > 0.05). Neither did median daytime PA intensity change (P > 0.05). At both registration periods, dogs spent approximately 80% (12 h) of their daytime hours sedentary, out of which just over 80% constituted the subfraction sleep-related sedentary behavior. There was a significant decrease of the subfraction non-sleep-related sedentary time, with a median difference of 2 min per day (P < 0.05). During the defined night, low levels of LMPA and VPA were recorded at both registration periods, with no significant changes (P > 0.05). Neither did median nighttime PA intensity change (P > 0.05). Dogs spent approximately 95% of nighttime hours sedentary at both registration periods, out of which 96% constituted the subfraction sleep-related sedentary behavior. There were some minor but significant changes

**Table 4. Accelerometer-derived patterns of physical activity behaviors and physical activity intensity in dogs (n = 15) during the defined day and night.**

| Defined day | | | |
|---|---|---|---|
| **Intensity category** | **Baseline, min per day** | **Intervention, min per day** | **P-value** |
| Sedentary behavior | 745 (607–805) | 742 (581–820) | 0.36 |
| Sleep-related sedentary behavior | 598 (447–709) | 628 (441–739) | 0.85 |
| Non-sleep-related sedentary behavior | 126 (70–179) | 124 (71–154) | *0.01* |
| LMPA | 185 (90–243) | 166 (97–268) | 0.33 |
| VPA | 24 (8–59) | 24 (8–70) | 0.56 |
| **PA intensity** | **Baseline, cpm per hour** | **Intervention, cpm per hour** | **P-value** |
| Vector magnitude | 871 (384–1341) | 719 (469–1475) | 0.85 |
| **Defined night** | | | |
| **Intensity category** | **Baseline, min per day** | **Intervention, min per day** | **P-value** |
| Sedentary behavior | 462 (436–470) | 458 (414–474) | 0.09 |
| Sleep-related sedentary behavior | 446 (413–459) | 435 (392–465) | *0.048* |
| Non-sleep-related sedentary behavior | 17 (9–25) | 22 (9–35) | *0.0006* |
| LMPA | 16 (10–34) | 18 (6–40) | 0.33 |
| VPA | 1.0 (0.2–10.0) | 1.6 (0.5–7.4) | 0.68 |
| **PA intensity** | **Baseline, cpm per hour** | **Intervention, cpm per hour** | **P-value** |
| Vector magnitude | 120 (74–408) | 149 (50–429) | 0.36 |

in the subfractions of nighttime sedentary behavior; median sleep-related sedentary behavior decreased by 11 min per night, while median non-sleep-related sedentary behavior increased by 5 min ($P < 0.05$). The mean length of sleep-like bouts in dogs was significantly longer during nighttime hours at both registration periods (approximately 20 min during nighttime compared to approximately 17 min during daytime, $P = 0.0001$).

Result from Wilcoxon matched-pairs signed rank test comparing baseline and intervention data. All data did not follow normal distribution, and results are presented as median and min–max values. Significant results are noted in italics. Daytime was defined as 07.00 a.m. to 22.59 p.m. and nighttime as 23.00 p.m. to 06.59 a.m. Sedentary behavior is presented both in total and divided into the subcategories sleep-related sedentary behavior and non-sleep-related sedentary behavior. Abbreviations: cpm, counts per minute; LMPA, light to moderate intensity physical activity; VPA, vigorous intensity physical activity

**Perceived physical capacity in dogs.** On the modified RPC-scale for dogs, dog owners registered a median of 4 (range 2–5) at baseline, representing being able to "trot at high speed for at least half an hour". At the intervention period, the corresponding figures were 4 (range 3–5), which was not a significant improvement ($P = 0.06$).

## Discussion

The main findings of this pilot study were a significant increase in VPA and a significant reduction in sedentary time in dog owners. There were no major changes in PA behaviors in dogs according to accelerometer registrations, but the slight decrease in BCS might still indicate positive effects related to the exercise program. Sleep-related data indicated satisfactory nighttime sleep patterns in both species, with no considerable effects of the program. Compared to accelerometer-derived data, dog owners underestimated both the time spent in MPA and in sedentary behavior. There was a high degree of fulfilment of recommendations for PA in dog owners, both at baseline and at the intervention period. However, both dog owners and dogs spent the main part of the daytime period sedentary, dogs mainly in the subfraction sleep-related sedentary behavior.

 

## Physical activity behaviors in dog owners

Accelerometer-derived patterns of physical behaviors in dog owners during awake hours were based on 24-h registrations, from which nighttime sleep periods were manually removed based on sleep detection in the software in combination with diary notations. Mean awake wear time was well above the minimum criterion of 10 h of daytime wear time (mean ± SD 16.0 ± 1.1 h at baseline and 15.0 ± 0.9 h at the intervention period). This may be regarded as a strength of the study, as short daily accelerometer wear time has been shown to result in underestimations of both sedentary and PA time [47]. Minimum wear time in days for inclusion in the study was based on established recommendations [32]. However, it cannot be excluded that longer registration periods could have captured the participants' physical activity behaviors in a more representative way.

Registrations during the intervention period showed an increase of VPA in dog owners of approximately 5 min per day, resulting in a total of 20 min of VPA per day (140 min per week) at the intervention period. This fraction of PA is of special importance for cardiorespiratory fitness, and thereby also for reducing morbidity and mortality [48–50]. VPA of merely 15–20 min per week has been associated with a 16–40% lower mortality hazard ratio, with further reductions up to 50–57 min per week [51]. In addition, there was a reduction of daily sedentary time by approximately 40 min. If this reduction is upheld on a longer basis, it might be of clinical relevance, as replacing 30 min of total sedentary time with PA has been associated with a significantly lower mortality risk [52]. Moreover, dog owners had high levels of MPA even at the outset of the study, which has been shown to attenuate, or even eliminate, the increased mortality risk associated with high sedentary time [49].

Accelerometer-derived data in dog owners showed high levels of fulfilment of WHO guidelines on PA. As the most recent guidelines recommend a weekly range of minutes of PA instead of merely a specified minimum and also emphasize the importance of higher levels of MVPA for people with high levels of sedentary time [14], we chose to assess the proportion of dog owners meeting both the lowest and highest criterion for each recommendation. Eleven dog owners fulfilled the highest achievable criterion of 300 min of MPA or MVPA per week even at baseline, and more than half also met the lowest achievable criterion of 75 min of VPA at that time point. This is in line with a growing body of evidence of a positive association between PA and dog ownership [5,21,53]. In comparison, it is estimated that less than one third of the world's population meet the lowest achievable criteria [15]. It is however possible that the dog owners who volunteered to participate in this pilot study were more physically active, or at least had a more positive attitude towards exercise, than the general population of dog owners in Sweden. Thus, the result may have been different with participants who were less physically active at the outset. Still, as only four participants selected target distances of 7.5 or 10 km, it might be plausible that even though the participants had high levels of MPA at baseline, they were less accustomed to performing more intense PA with their dogs. As VPA is of special importance for cardiovascular health [48,49], measures to increase this category of PA is valuable even for people with high levels of MPA.

The underestimations of PA in this study are contradictory to previous findings that self-reported measures of PA are generally higher than those directly measured by accelerometers [54]. This might reflect an integration of dog walking into the owner's lifestyle to such a degree that it's not thought of as PA. A qualitative study conducted by Westgarth et al. (2017) [55] found that dog owners primarily reported feelings of happiness in conjunction with dog walking, while PA was merely regarded as a secondary bonus. However, apart from underestimations of PA, the dog owners in this study also underestimated their sedentary time at both registration periods. According to accelerometer-derived data, most of the daily wear time of dog owners was spent sedentary. This is in concordance with previous research on the general population [24,56,57]. Accelerometer-derived sedentary time in dog owners in specific is however less well studied, with partly contradictive results [58–60]. The results from this study indicate that the dog owners did in fact not have a more beneficial baseline pattern of sedentary behavior than the general Swedish population [24]. Sedentary time exceeding 9.5 h per day has been associated with a significantly higher risk of death [2]. At baseline, 10 out of 15 dog owners exceeded this limit. This finding is not unexpected, as individuals in predominantly seated occupations may be likely to accumulate up to

eight hours or more of sedentary time during working hours alone. The underestimations of sedentary time, in combination with the relatively high levels as measured by accelerometry, highlight the importance of including not only PA but also sedentary behavior when studying activity levels or performing PA interventions in dog owners, as a sedentary lifestyle repeatedly has been associated with increased risk for several major chronic disease outcomes [61,62].

There were no significant changes in self-perceived physical capacity in dog owners as assessed by the PRC-scale; the participants assessed that they were able to jog for at least half an hour at both registration periods. Here, it needs to be considered that most dog owners selected the very shortest target distances (2 or 5 km), indicating no or little previous jogging experience. This may be of relevance as it has previously been shown that untrained healthy adults demonstrated a tendency to overestimate their actual physical capacity [63]. It might thus be reasonable to hypothesize that the assessments made by the dog owners at the end of the intervention more accurately reflected their actual physical capacity than their baseline assessments.

Dog owner sleep data derived from accelerometry and questionnaires indicated adequate levels of sleep [64] and no, or low, levels of problems related to insomnia, sleep quality and daytime sleepiness on cohort level at both registration periods, which may be related to the high levels of PA in general [4]. Most dog owners stated that they never or rarely woke up during the night because of the dog, which is consistent with the low nighttime activity levels measured by accelerometry in participating dogs. Human-dog co-sleeping has previously been studied with contradictory results [28,65–67], indicating that this phenomenon needs to be investigated further in order for health care workers to give evidence-based recommendations on the potential value of dog ownership on sleep.

There were no changes in BMI or WHR in dog owners, which is in line with findings from previous exercise interventions without dietary restrictions [68,69]. There are, however, contrasting studies that do in fact show significant reductions of BMI and other body measurements as a result of PA only [70–72], but these studies are generally based on exercise interventions that lasted longer than eight weeks. Moreover, the fact that a substantial proportion of dog owners selected the shortest target distances of 2 or 5 km may also have prevented changes on cohort level, as the workload may not have been sufficient to cause significant reductions in BMI or body measurements in this short period of time. In addition, as no dietary instructions were given to the participants, it cannot be excluded that the dog owners might in fact have increased their food intake during the intervention if their appetite increased in response to a higher exercise load. In this respect, it should be noted that the aim of this specific joint exercise program, as stated by its creator the Swedish Working Dog Association, is not specifically to reduce weight in either dog owners or dogs but rather to increase outdoor physical activity, thereby improving health and wellbeing in both parties [29]. If weight loss is desirable, the exercise program should ideally be combined with appropriate dietary instructions and upheld for longer periods of time.

### Physical activity behaviors in dogs

Accelerometer-derived patterns of PA behaviors in dogs were based on 24-h registrations, separated into predefined day- and nighttime periods. Although this division of data is unlikely to capture the exact diurnal rhythm of all participating dogs, it enabled crude distinctions between daytime and nighttime behavioral patterns. Median wear time was high; close to 24 h at both registration periods. This may be regarded as a strength, as high levels of accelerometer wear time are likely to result in more correct estimations of physical behaviors [47]. As for dog owners, established recommendations for minimum wear time in days [33] were adhered to in the study, but longer registration periods may have captured the dogs' PA behaviors in a more representative way.

Dogs spent more than 12 of the 16 h defined as day sedentary at both registration periods, leaving less than 4 h for PA of different categories. Most of these sedentary hours constituted sleep-related sedentary behavior, indicating very low activity levels. There was a small but significant decrease in the subfraction non-sleep-related sedentary behavior, but as the distinction between day- and nighttime in this study was based on an arbitrary division for practical reasons, this minor decrease might be a chance finding without clinical relevance. Although direct comparisons should be avoided due

to differences in accelerometer settings and definitions of daytime, the high levels of daytime sedentary behavior and relatively low levels of PA registered in this study are line with previous research in companion dogs [7,73,74].

According to the accelerometer registrations, there was no increase in PA in dogs at the intervention registration period compared to baseline. Whether this finding should be interpreted as indicating that the exercise intervention was insufficient to increase PA in dogs or whether alternative explanations may exist warrants further consideration. On the one hand, it is possible that jogging with the owner was not enough to increase PA levels in dogs on cohort level. It may be reasonable to assume that the dog owners replaced some of their regular dog walks (including periods of off leash-running) with the jogging sessions, rather than keeping their regular walking habits and adding the exercise sessions on top of that. Since most dog owners chose the shorter target distances of 2 or 5 km, it is also reasonable to assume that they were rather unexperienced joggers and that their pace during the jogging sessions was not remarkably high. Thus, the dogs may very well have been able to trot rather than run beside their owners even during the jogging sessions, which then would not have generated any additional time in VPA for dogs. Since the established cut-offs for accelerometry for dogs do not distinguish between walking and trotting but rather classifies them both as LMPA [7], it is unknown if their time in trotting increased on behalf of the time spent walking. However, the fact that their median PA intensity did not change argues against that.

On the other hand, it is possible that PA levels in dogs did increase but were not detected due to methodological constraints, such as insufficient length of the registration periods. The intervention registration period constituted a small sample of the eight-week exercise program and may not have been enough to capture a representative PA pattern for participating dogs during the final week. In addition, it cannot be excluded that the dogs were in fact exercised a bit extra compared to their regular habits when the accelerometers were first put on at baseline, as a form of eagerness to please-bias from the dog owners. In that case, the baseline registrations might have been somewhat exaggerated, and the registrations during the intervention period might in fact represent an increase in PA compared to their true regular habits. Furthermore, as opposed to the development of age-specific accelerometer-settings and processing criteria in humans, such as different cut-offs for children compared to adults, there is yet no equivalent advancement for accelerometer-based PA measurements in dogs despite their diversity in body size and conformation. However, the established cut-offs for PA intensity categories for dogs in general [7] have been shown to be valid and reliable for assessing PA volume and intensity across a range of dog breeds, ages and sizes [26]. Still, breed and age may have influenced the outcomes of the exercise program, as these factors may affect an individual's capacity for PA. Groupwise analyses were however not possible due to the small sample size and high diversity in breed and age.

Furthermore, the lack of detected changes in PA in dogs needs to be discussed in relation to the small, but still significant, reduction in BCS that was registered after an exercise intervention lasting only eight weeks. In 10/15 participating dogs, BCS was reduced by one point on the 9-point BCS scale. This scale has been shown to be an excellent tool for assessing BCS in dogs, as Laflamme (1997) [46] demonstrated a highly significant correlation between BCS and body fat percentage as determined by dual energy X-ray absorptiometry (DEXA). Additionally, Laflamme (1997) showed assessments to be repeatable both within and between assessors [46]. Still, as assessments are made based on visual and palpatory hallmarks of body fat content there is always a risk for some degree of subjectivity. To minimize the risk of assessor-dependent variations, the same veterinarian with specific expertise conducted all BCS assessments at both registration periods in this pilot study. As discussed above, no dietary restrictions were given to participating dog-dog owners pairs. The fact that dogs generally do not have the possibility to freely adjust their food intake according to their level of PA might have contributed to the fact that a slight reduction in BCS was registered, while there was no reduction in owner BMI. Furthermore, a previously published paper based on the same intervention but with a larger cohort of participants [75] showed a significant increase in the dogs' thigh circumference and a significant decrease in chest circumference while their bodyweight was stable, indicating a redistribution between body fat and muscle mass. Taken together, this suggests that there may have been positive changes in the PA patterns of dogs during the intervention, even though this was not detected in the accelerometer registrations.

Dogs spent the absolute majority of defined nighttime hours in sleep-related sedentary behavior at both registration periods. Here, it needs to be stressed that the definition of sleep-related sedentary behavior in this study is inspired by studies by Ladha and Hoffman et al. (2018) [27] and Hoffman (2020) [28], in which they found that dogs' movements below 150 cpm were so subtle that they were likely to be either asleep or in a sleep-like state. This should however not be equated with the use of a validated sleep algorithm in humans. Still, it may be argued that the use of a cut-off to distinguish the very lowest levels of canine sedentary behavior is of value in this context, as the interval for sedentary behavior in dogs is markedly broader than the corresponding for humans (0–1352 cpm in dogs versus 0–199 cpm in humans [7,35]). There were no significant changes in nighttime activity behaviors in dogs, except for a minor shift in the proportion of sleep-related and non-sleep-related sedentary behavior. This may be a chance finding without clinical relevance, as the distinction between day- and nighttime in this study is an arbitrary division for practical reasons. The low, but still existing, levels of VPA registered during defined nighttime hours represented early morning habits in a few dog-dog owner pairs and should thus not be interpreted as nighttime movements of intense character. Rather, the predominant occurrence of sleep-related sedentary behavior during the defined nighttime suggests that this cohort of dogs did in fact spend most of these hours in sleep or a sleep-like state. To explore the characteristics of the polyphasic nature of canine sleep during the intervention, sleep-like bouts with a defined duration of ≥ 10 min in sleep-related sedentary behavior were used. The mean length of sleep-like bouts was significantly longer during nighttime hours compared to daytime at both registration periods; approximately 20 min compared to 17 min. More studies are needed to confirm these findings, and to assess their possible clinical relevance. Adams and Johnson (1993) [76] noted an average length of nighttime sleep periods of 16 min. Direct comparisons should however be avoided, as Adams and Johnson (1993) [76] collected their data through observation of data recordings of the dogs and not through accelerometry.

## Limitations and future directions

Results from this pilot study should be regarded with some caution, primarily due to methodological constraints. First, the limited sample size may have increased the risk of type II-error, potentially masking positive changes in outcome measures. Still, the fact that significant changes in dog owners' PA behaviors were observed despite the limited sample size suggests that joint dog-dog owner exercise is a phenomenon that warrants further research. There was a relatively large drop-out during the study, mostly for unknown reasons. To further assess the feasibility of joint dog-dog owner exercise programs, reasons for drop out should have been investigated in a more in-depth manner. Second, even though established recommendations for minimum accelerometer wear time were adhered to in the study, longer registration periods may have captured patterns of PA in both species in a more representative way. Third, participating dog owners and dogs acted as their own controls. In future studies of joint exercise interventions, it is recommended to include dog-dog owner pairs who continue with their regular PA habits, and to randomize participants to either active intervention or control group. Fourth, there was no long-term follow-up. This should be included in future studies, as increases in PA and reductions of sedentary time need to be upheld on long-term basis to give lasting health effects. Moreover, procedures for accelerometer data collection and processing in dogs are still in their early stages of development compared to those for humans. As there are no established canine wear time validation parameters or validated sleep algorithms in dogs, accelerometer-derived patterns of PA behaviors in participating dogs should be regarded more guardedly than corresponding results from dog owners. Furthermore, the value of the dog as a training partner should be explored to evaluate the possible additional value of performing an exercise intervention together with one's dog instead of on your own or with a human training partner. This could be achieved through validated human-dog bond questionnaires in combination with qualitative inquiry. Relatively few and mild injuries were reported by the participants. However, it should be noted that baseline BMI in dog owners ranged from normal weight to class 1 obesity (BMI 18.5–<35), and BCS scores in dogs ranged from normal weight to overweight (BCS 4–7). For dog owners and dogs with more severe overweight or obesity, initial dietary weight reduction may be wise to recommend to avoid harmful strain from jogging on joints and ligaments.

## Conclusions

The results from this pilot study indicated that the joint outdoor exercise program can be useful for increasing vigorous physical activity and reducing sedentary time in dog owners, which is of importance both for maintaining cardiovascular health and for reducing the risk for non-communicable diseases. No major changes in physical activity behaviors in dogs were registered, but the slight decrease in body conditions score might indicate positive effects related to the exercise program. Sleep-related data indicated satisfactory nighttime sleep patterns in both dogs and owners with no considerable effects of the program. The high degree of baseline fulfilment of recommendations for physical activity supports previous findings of high physical activity levels in dog owners, and the underestimation of self-reported moderate-intensity physical activity suggests a strong integration of dog walking habits into daily routines. However, the high levels of sedentary time underline the importance of assessing sedentary behavior in addition to physical activity in both species in future studies. The results of this study should be confirmed in randomized controlled studies with larger sample size and long-term follow-up.

## Supporting information

**S1 File. Demographics, physical activity behavior data, body measurements and questionnaires.**
(XLSX)

**S1 Table. Accelerometer-derived sleep-related behaviors in dog owners (n = 15).** Results from Wilcoxon matched-pairs signed rank test comparing baseline and intervention data. All data were not normally distributed, and results are presented as median and min–max values. Significant results are noted in italics.
(DOCX)

**S2 Table. Questionnaire-derived sleep-related behaviors in dog owners (n = 15).** Results from Wilcoxon matched-pairs signed rank test comparing baseline and intervention data. All data were not normally distributed, and results are presented as median and min–max values.
(DOCX)

**S1 Image. Striking image.**
(TIF)

## Acknowledgments

We thank all participating dog owners and dogs and Britta Agardh from The Swedish Working Dog Association for their contribution to this study.

## Author contributions

**Conceptualization:** Anna Bergh, Erika Roman.

**Data curation:** Klara Smedberg.

**Formal analysis:** Klara Smedberg.

**Funding acquisition:** Anna Bergh, Erika Roman, Jan W. Eriksson, Josefin Söder.

**Investigation:** Anna Bergh, Erika Roman, Sören Spörndly-Nees, Lena V. Kallings, Josefin Söder.

**Methodology:** Anna Bergh, Erika Roman, Sören Spörndly-Nees, Jan W. Eriksson, Lena V. Kallings, Josefin Söder.

**Project administration:** Anna Bergh, Erika Roman.

**Supervision:** Anna Bergh, Erika Roman, Sören Spörndly-Nees, Jan W. Eriksson, Josefin Söder.

**Validation:** Klara Smedberg.

**Visualization:** Klara Smedberg.

**Writing – original draft:** Klara Smedberg.

**Writing – review & editing:** Klara Smedberg, Anna Bergh, Erika Roman, Sören Spörndly-Nees, Jan W. Eriksson, Lena V. Kallings, Josefin Söder.

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
