## [Decision Letter · Decision Letter 0]

20 Jan 2026

Dear Dr. Smedberg,

We look forward to receiving your revised manuscript.

Kind regards,

Wolfgang Blenau

Academic Editor

PLOS One

Journal Requirements:

“The salary of K.S. was funded by the Uppsala Diabetes Centre (UDC), Sweden (https://www.uu.se/centrum/diabetes). J.S. received research grants from the “Research Foundation of Agria and the Swedish Kennel Club” (https://www.skk.se/agria-och-skk-forskningsfond/) and A.B. received research grants from “SLU Future One Health”, Swedish University of Agricultural Sciences (https://www.slu.se/en/about-slu/organisation/future-platforms/slu-future-one-health/). J.W.E. and E.R. had funding from the Swedish Diabetes Foundation (https://www.diabetes.se/forskning/diabetesfonden/), the Swedish Research Council (https://www.vr.se/english.html, project 2024-03344) and the Swedish Foundation for Strategic Research (https://strategiska.se/en/, project CMP22-0014). J.W.E. also had funding from the Novo Nordisk Foundation (https://novonordiskfonden.dk/en/) and the European Commission Horizon RIA project PAS GRAS (https://pasgras.eu/en/about-pas-gras, grant agreement ID 101080329).”

3. We note that there is identifying data in the Supporting Information file <file name>. Due to the inclusion of these potentially identifying data, we have removed this file from your file inventory. Prior to sharing human research participant data, authors should consult with an ethics committee to ensure data are shared in accordance with participant consent and all applicable local laws.

-Location data

4. We noted in your submission details that a portion of your manuscript may have been presented or published elsewhere. “Yes. This manuscript is based on unique valid accelerometer data and unique data from sleep registrations and diaries from dog owners and their dogs. This data has not been published previously.

Data on body measurements and some questionnaire-based data on physical activity/sedentary time included in the manuscript have previously been published elsewhere (please see related manuscripts). The previously published data in this manuscript is merely included for comparisons of direct accelerometer-derived data (unique data in this manuscript) and self-assessed data, or presented as secondary outcomes of physical activity.

The previous publications were based on a larger cohort with other research questions, aims and results. The unique data presented in this manuscript are derived from a subgroup from the larger cohort. Valid accelerometer data and valid data from sleep registrations and diaries were not provided for all participants in the larger cohort, and merging of manuscripts were not possible. Therefore, the inclusion and use of data in this submission does not constitute dual publication.” Please clarify whether this [conference proceeding or publication] was peer-reviewed and formally published. If this work was previously peer-reviewed and published, in the cover letter please provide the reason that this work does not constitute dual publication and should be included in the current manuscript.

5. We note that you have indicated that there are restrictions to data sharing for this study. PLOS only allows data to be available upon request if there are legal or ethical restrictions on sharing data publicly. For more information on unacceptable data access restrictions, please see http://journals.plos.org/plosone/s/data-availability#loc-unacceptable-data-access-restrictions.

Reviewers' comments:

Reviewer's Responses to Questions

**Comments to the Author**

1. Is the manuscript technically sound, and do the data support the conclusions?

Reviewer #1: Yes

Reviewer #2: Partly

2. Has the statistical analysis been performed appropriately and rigorously?

Reviewer #1: Yes

Reviewer #2: N/A

3. Have the authors made all data underlying the findings in their manuscript fully available?

Reviewer #1: Yes

Reviewer #2: Yes

4. Is the manuscript presented in an intelligible fashion and written in standard English?

Reviewer #1: Yes

Reviewer #2: Yes

Reviewer #1: This is a great pilot study to look at the effects of human-animal joint exercise on important health outcomes like BMI and BCS. It is well written and results are clear. There are a number of limitations, as there are for many pilot studies, which the authors do address most of them, however, I have suggested a few others to consider.

Abstract

- It would be useful to provide a sentence on what kind of exercise regimen you implemented, i.e. describe the methods of the study before listing the results

- Provide data values and p-values for the results you are listing

Introduction

Line 47: Remove “since decades”

Line 54: You have defined physical activity as PA so use the abbreviation throughout the manuscript

Line 55-57: Replace the instances of “associated to” with “associated with”

Methods

Line 76: why were the dogs and humans instructed to wear the accelerometers for different lengths of time (humans 4 days and dog 3 days)?

Line 222-225: This sentence is unclear. Why did you multiply by 7 when you had distinct data for 4 separate days? What does 150 respectively 300 min mean? Why did you choose to make each time a criterion when the WHO recommends any time within this range? Why did you choose the minimum and maximum?

Line 279: Should be “nearest mm and kg, respectively.”

Results

Line 319: So if your total sample size is only 13-15, perhaps a more conservative test of normality should be used like Shapiro-Wilk?

Line 360: It should be 15 and 17%, respectively. I would recheck the entire manuscript for this as I believe you use the term ‘respectively’ quite often.

Discussion

Given that the exercise routine that was implemented involved increasing distance and time each week yet the dogs physical exercise did not increase from baseline, do you think this means that the owners did not comply with the exercise regimen? The dogs physical activity should have increased unless they were all already running their end goal distance at baseline right? Especially since this was only measured once at the end where maximal exercise should have been achieved.

Do you think your population of humans was generally more physically active given that a lot of them met the WHO criteria even at baseline. If so, this should be called out as your results may be totally different had you recruited people who were more inactive. Similarly, the humans used were generally not overweight so this may explain why you saw no differences in BMI or WHR. Did you look at the data based on baseline BMI? Maybe those who started with a greater BMI did show decreases by the end?

I think you are missing a big piece of information for the dogs especially but also for the humans and that is diet. I would suspect that even though the dogs were running more as part of the exercise regimen, the human was not adjusting their food intake whereas if the human was doing more exercise, they may be hungrier and eat more for themselves. I think diet needs to be addressed in your discussion relating to the change in BCS but not change in BMI. On the contrary, it could also be explained by the subjectiveness of a BCS in dogs – maybe the change observed was just due to error. Theoretically if the dogs exercise did not change according to your results then their BCS should not change unless the owners were restricting the dogs feed intake which you do not have data on.

Reviewer #2: The study presents considerable potential and value for veterinary science, particularly by addressing physical activity and body condition in dogs alongside their owners. I would also like to highlight that the connection established with the One Health concept is essential in elevating the study to a public health perspective, demonstrating its relevance not only to veterinary science or human health sciences, but to both in an integrated manner. In my view, the topic of canine body condition remains underexplored and insufficiently discussed, particularly in my continent, as observed during my own research in this field. Furthermore, assessing physical activity in association with sleep improvement represents an essential perspective that deserves further exploration within veterinary research.

I have several additional points to raise during my evaluation; however, it is important to note that most of these have already been acknowledged and, in some cases, explicitly discussed by the author, who recognises several limitations of the study.

Firstly, I consider the article to be methodologically constrained, primarily due to the small sample size. I acknowledge that conducting a study of this scope, involving multiple assessments that are highly dependent on participant compliance, requires a substantial level of commitment from dog owners throughout the entire project. Nevertheless, the limited sample size may introduce significant bias across the statistical analyses, potentially masking positive findings that may have been observed but did not reach statistical significance (p > 0.05). Although the challenges in maintaining a larger sample are understandable and clearly justified by the author, this limitation reduces the robustness of the results and highlights the need for further studies with larger samples to allow more definitive conclusions.

Another point of concern relates to the eight-week evaluation period, which raises questions regarding whether meaningful changes could realistically occur within such a short timeframe and be detected statistically. Given the relevance and alignment of the study objectives, a longer follow-up period (such as six months or even one year) might be more appropriate to capture more definitive outcomes. While extended follow-up presents practical challenges due to participant adherence, this remains an important consideration.

Additionally, the structure of the baseline and intervention periods warrants further clarification. It is unclear why accelerometers were not used throughout the entire eight-week period, as continuous monitoring would have generated a more comprehensive dataset for statistical analysis. Moreover, a single week used as a reference period appears insufficient, as it may represent an atypical week in the routine of the owner and dog. This concern similarly applies to the intervention period, as data appear to have been collected only during the final week of the study. Short assessment windows increase the likelihood of outliers and introduce additional bias. Extending data collection over a longer period would likely provide a more reliable representation of habitual behaviour. Still on the subject of this time period, it was confusing to my understanding: the study lasted eight weeks, but only two weeks of data collection. What were the other six weeks used for?

Regarding the outdoor exercise programme, allowing owners to choose their preferred distance is a positive aspect; however, this approach may also have influenced the statistical outcomes. The absence of significant changes in secondary outcomes, such as BMI and BCS, may be partially attributable to the fact that a substantial proportion of participants selected shorter distances. A discussion or stratified analysis comparing the 2, 5, 7.5 and 10 km groups could have provided valuable insights into the relationship between exercise distance, physical activity levels, sedentary behaviour, sleep patterns, and secondary measures. While subdividing an already small sample presents limitations, the influence of distance travelled warrants consideration.

The author notes that exercise distance and intensity were progressively increased throughout the programme; however, it is unclear whether this refers to running sessions, circuit training, or both. Providing further clarification would reduce potential misinterpretation.

Another issue that creates a bias in the assessment: the days of accelerometer use were not standardised. To what extent could this variation (even if only one to two days at most) also influence the final results?

With respect to physical activity classification, although WHO guidelines were applied, these classifications rely partly on self-report. As acknowledged by the author, owners tended to underestimate their physical activity levels, introducing subjective bias. Future studies might benefit from prioritising objective measures, such as accelerometer counts per minute, supplemented by owner reports when necessary.

The assessment of sedentary behaviour also appears insufficiently described. For owners engaged in predominantly seated occupations, it is unclear how periods of low counts per minute were classified. Similarly, for dogs whose owners work outside the home full-time, opportunities for physical activity may be limited unless deliberately facilitated. Addressing these aspects would strengthen the interpretation of sedentary behaviour.

Now about the samples, I have some questions about dog owners. First, I think it would be important to highlight and discuss the influence of age on physical activity, because as we age, we humans naturally decrease our exercise and even sleep time. Perhaps the research group could assess whether there were few changes in older humans, if this may have masked the results as well. Finally, regarding the owners, during their mental health assessment, were anxiety or depressive disorders taken into account, for example? It is also known that these are common in the human population and directly affect the ability and amount of exercise. Now, regarding the dogs, we must also consider the issue of breed and age. Some breeds have greater ‘energy’ and exercise more easily because they are animals genetically bred for hunting, for example, or with better biomechanics for exercise. In addition, younger animals are also more predisposed to exercise, and this may also influence the results. Again, I understand that the sample was small, preventing the authors from making such assessments, but it is my job to point out this bias and suggest different assessments to be made in a future study (or even discussions to be held in this article).

Now, regarding the assessment of dogs' sleep, using whether or not the owner was awakened by their pet during the night as a basis for assessing the dogs' sleep is of little use, as this does not mean that the dog was having a proper, uninterrupted period of sleep, only that it did not go to its owner or that the owner did not wake up with their pet.

Regarding the BCS, how was it performed? Did a single assessor perform the entire BCS for each patient during the reference period and after the intervention period? Was there more than one assessor? Was the assessor experienced? I believe this should be reported because, as we know, although the BCS is an excellent indicator of a dog's body condition, it has a significant flaw in terms of subjectivity, as its qualitative nature can lead to assessor-dependent variations. Therefore, if different assessors were involved during the study or if they were inexperienced, this could explain some changes or even the absence of changes. In addition, the author could discuss precisely this point: that despite the minimal variations in BCS in the study, it is a subjective parameter and that in a short period of time it may not show visible and significant variations to the extent that the assessor can identify a change in score.

I conclude here with praise, as the study involves a topic that is rarely discussed but deserves closer attention from researchers. Even with the small sample size, it managed to overcome this difficulty by producing an excellent article with an impeccable discussion. That said, I would like to add that the discussion presented by the authors is extremely satisfying to read, follows a fluid line of reasoning, and exposes all the difficulties and possible biases that the work faced, without neglecting the importance of the research. The authors themselves highlight most of the limitations I describe here (albeit with some flaws to be corrected or improved upon, at least in the future) and also point out the lack of research on the subject, which prevents some more in-depth discussions. However, I am pleased to note that this study has begun the process of generating measures to establish future standards for these accelerometer values, physical activities, and sleep assessments in dogs, and even more so that all of this is intertwined (very well thought out) with the concept of One Health in a vision focused on human and animal health in unison.

.

Reviewer #1: No

Reviewer #2: **Yes:**Ricardo de Souza BuzoRicardo de Souza BuzoRicardo de Souza BuzoRicardo de Souza Buzo

---

## [Author Response · Author response to Decision Letter 1]

3 Mar 2026

Point-by-point answers to the Editor

Journal requirements

This has been addressed.

2. Please state what role the funders took in the study.

3. We note that there is identifying data in the Supporting Information file <file name>. Due to the inclusion of these potentially identifying data, we have removed this file from your file inventory. Please remove or anonymize all personal information (<specific identifying information in file to be removed>), ensure that the data shared are in accordance with participant consent, and re-upload a fully anonymized data set.

To adhere to the ethical permit and not compromise the privacy of participating dog owners, dog owner raw data is provided in de-identified form. We have uploaded a fully de-identified data set (S1_File). There are no ethical or legal restrictions on sharing this de-identified data set.

4. We noted in your submission details that a portion of your manuscript may have been presented or published elsewhere. Please clarify whether this [conference proceeding or publication] was peer-reviewed and formally published. If this work was previously peer-reviewed and published, in the cover letter please provide the reason that this work does not constitute dual publication and should be included in the current manuscript.

This manuscript is primarily based on accelerometer-derived data, which is unique and has never been published before. Data on body measurements and some questionnaire-derived data on physical activity/sedentary time included in this manuscript have previously been peer reviewed and formally published elsewhere, but for a larger cohort and with other research questions, aims and results (please see related articles: Smedberg K, Lundbeck E, Roman E, Eriksson JW, Spörndly-Nees S, Kallings LV, et al. A pilot study of a joint outdoor exercise program for dog owners and dogs. Sci Rep. 2024;14(1). doi: 10.1038/s41598-024-65033-0., Söder J, Roman E, Berndtsson J, Lindroth K, Bergh A. Effects of a physical exercise programme on bodyweight, body condition score and chest, abdominal and thigh circumferences in dogs. BMC Vet Res. 2024;20(1). doi: 10.1186/s12917-024-04135-3.). The rationale for including these data in the current manuscript was to fulfill the following aims: 1) to enable comparisons between physical activity/sedentary time as measured by accelerometry, questionnaires and activity diaries, and 2) to assess possible effects of accelerometer-derived physical activity on body measurements. As valid accelerometer and diary data were not provided for all participants in the larger cohort, these aims could only be used in the current study and its smaller cohort of participants. As such, we do not consider the manuscripts to be dual publications.

5. We note that you have indicated that there are restrictions to data sharing for this study. PLOS only allows data to be available upon request if there are legal or ethical restrictions on sharing data publicly.

The authors confirm that the data supporting the findings of this study are available within the article and its supplementary material. To adhere to the ethical permit and not compromise the privacy of participating dog owners, dog owner raw data is provided in de-identified form. We have uploaded a fully de-identified data set (S1_File). There are no ethical or legal restrictions on sharing this de-identified data set.

N/A.

7. Please review your reference list to ensure that it is complete and correct. Any changes to the reference list should be mentioned in the rebuttal letter that accompanies your revised manuscript.

Please note that two references have been added to the manuscript: Söder J, Roman E, Berndtsson J, Lindroth K, Bergh A. Effects of a physical exercise programme on bodyweight, body condition score and chest, abdominal and thigh circumferences in dogs. BMC Vet Res. 2024;20(1). doi: 10.1186/s12917-024-04135-3 (refence no. 74), Ekelund U, Tarp J, Ding D, Sanchez-Lastra MA, Dalene KE, Anderssen SA, et al. Deaths potentially averted by small changes in physical activity and sedentary time: an individual participant data meta-analysis of prospective cohort studies. Lancet. 2026;407(10526):339–49. doi: 10.1016/s0140-6736(25)02219-6.(reference no. 50).

Point-by-point answers to the Reviewer comments

Response to Reviewer 1

We are sincerely grateful for your relevant feedback and constructive suggestions that really have guided us to improve the manuscript.

Abstract

- It would be useful to provide a sentence on what kind of exercise regimen you implemented, i.e. describe the methods of the study before listing the results

- Provide data values and p-values for the results you are listing

Thank you, these comments have been addressed.

Introduction

Line 47: Remove “since decades”

Line 54: You have defined physical activity as PA so use the abbreviation throughout the manuscript

Line 55-57: Replace the instances of “associated to” with “associated with”

Thank you, these comments have been addressed throughout the revised manuscript.

Methods

Line 76: why were the dogs and humans instructed to wear the accelerometers for different lengths of time (humans 4 days and dog 3 days)?

Thank you for pointing out the need for clarification. They were instructed to wear the accelerometer during one week at baseline and one week at the intervention period, but only participants with at least 4 days (humans) or 3 days (dogs) of valid data per period were included in the study. These specific periods of days were based on previous research in each species. This has now been clarified in the revised manuscript.

Line 222-225: This sentence is unclear. Why did you multiply by 7 when you had distinct data for 4 separate days? What does 150 respectively 300 min mean? Why did you choose to make each time a criterion when the WHO recommends any time within this range? Why did you choose the minimum and maximum?

Since the participants had valid data for less than a week per registration period, we chose to multiply their daily average by seven to enable assessment of fulfilments of WHO recommendations of weekly physical activity time. As the most recent guidelines recommend a weekly range of minutes of PA instead of merely a specified minimum and emphasize the importance of higher levels of MVPA for people with high levels of sedentary time, we chose to assess the proportion of dog owners meeting both the lowest and highest criteria for each recommendation. The sections regarding WHO recommendations have now been revised and clarified in the method section and in the discussion.

Line 279: Should be “nearest mm and kg, respectively.”

Thank you, this has been addressed.

Results

Line 319: So if your total sample size is only 13-15, perhaps a more conservative test of normality should be used like Shapiro-Wilk?

Thank you for this valuable suggestion. We have re-analyzed our data with Shapiro-Wilk’s test, which did indeed result in some changes in the distribution of data and therefore also to some changes of method of analysis for detecting changes between registration periods (from t-test to Wilcoxon). Two results that were significant in the first version of the manuscript did not reach statistical significance (P > 0.05), and these were: 1) There was no longer a significant increase in owner-rated perceived physical capacity in dogs (P = 0.06 with Wilcoxon vs 0.03 when t-test was used). 2) When comparing questionnaire-derived PA data with accelerometry there was no longer a significant underestimation of VPA at baseline (P = 0.06 vs 0.03 when t-test was used). We have now revised the relevant sections of the manuscript accordingly.

Line 360: It should be 15 and 17%, respectively. I would recheck the entire manuscript for this as I believe you use the term ‘respectively’ quite often.

Thank you, this has been addressed.

Discussion

Given that the exercise routine that was implemented involved increasing distance and time each week yet the dogs physical exercise did not increase from baseline, do you think this means that the owners did not comply with the exercise regimen? The dogs physical activity should have increased unless they were all already running their end goal distance at baseline right? Especially since this was only measured once at the end where maximal exercise should have been achieved.

Thank you for this interesting question, which we also have spent quite some time thinking about when analyzing the data and drafting the manuscript. Based on data from the dog owner questionnaires and activity diaries, it seems reasonable to believe that the vast majority of dog owners did comply with the exercise regimen (although two participants stated that they were not able to reach their selected target distance of 2 km). So, the question is, was there no increase in dogs’ activity levels, or was there an increase that we failed to capture? There may be good arguments for both alternatives. First, it may be reasonable to guess that the dog owners replaced some of their regular dog walks with the jogging sessions, rather than keeping their regular walking habits and adding the exercise sessions on top of that. In Sweden, dogs may be kept off leash under supervision, and most Swedes have access to greenspaces in their local area, which allows time for regular off-leash running for companion dogs. If time in off-leash running was a part of the dogs’ regular walks, exchanging walks for jogging sessions with the owner might not automatically be beneficial for VPA levels in dogs. Furthermore, since most dog owners chose the shortest target distances of 2 or 5 km, it is likely to assume that they were rather unexperienced joggers and that their pace during the jogging sessions was not remarkably high. Thus, the dogs may very well have been able to trot rather than run beside their owners even during the jogging sessions, which then would not have generated any additional time in VPA for dogs. Since the established cut-offs for accelerometry for dogs do not distinguish between walking and trotting, but rather classify them both as LMPA, we do not know if their time in trotting increased on behalf of the time spent walking. However, the fact that their median PA intensity did not change argues against that.

On the other hand, it may be possible that PA levels in dogs did in fact increase but were not detected due to methodological constraints, such as insufficient length of the registration periods. The intervention registration period constituted a small sample of the eight-week exercise program and may not have been enough to capture a representative PA pattern for participating dogs during the final week. In addition, it cannot be excluded that the dogs were in fact exercised a bit extra compared to their regular habits when the accelerometers were put on at baseline. In that case, the baseline registrations might have been somewhat exaggerated, and the registrations during the intervention period might in fact represent an increase in PA compared to their true regular habits. Furthermore, the small, but still significant, reduction in BCS (please see further discussion on BCS below) suggests that perhaps something did change in the dogs’ physical activity patterns during the study. To conclude, more studies are needed to evaluate the effect of this exercise program on PA levels in dogs. These should be conducted as randomized controlled trials, using larger cohorts and preferably longer registration periods. We have now tried to elaborate our line of reasoning regarding the lack of changes in PA in dogs in the revised manuscript.

Do you think your population of humans was generally more physically active given that a lot of them met the WHO criteria even at baseline. If so, this should be called out as your results may be totally different had you recruited people who were more inactive.

Yes, it is indeed plausible that our study group were more physically active than the Swedish dog owner population in general, and that they had a more positive attitude towards exercise as they volunteered to take part in the intervention. This might very well have affected our results, which we now have clarified in the revised manuscript.

Similarly, the humans used were generally not overweight so this may explain why you saw no differences in BMI or WHR. Did you look at the data based on baseline BMI? Maybe those who started with a greater BMI did show decreases by the end?

On a cohort level, the dog owners had a median BMI of 26.2 at baseline, representing only slight overweight. However, looking at the individual levels, 6/15 dog owners were obese at baseline, and 5/15 at the intervention registration period. Slight decreases in BMI were registered in 10/15 participants, but we did not see greater decreases in more obese subjects. We have now specified the information regarding BMI data in the revised manuscript.

I think you are missing a big piece of information for the dogs especially but also for the humans and that is diet. I would suspect that even though the dogs were running more as part of the exercise regimen, the human was not adjusting their food intake whereas if the human was doing more exercise, they may be hungrier and eat more for themselves. I think diet needs to be addressed in your discussion relating to the change in BCS but not change in BMI. On the contrary, it could also be explained by the subjectiveness of a BCS in dogs – maybe the change observed was just due to error. Theoretically if the dogs exercise did not change according to your results then their BCS should not change unless the owners were restricting the dogs feed intake which you do not have data on.

Thank you, you do absolutely have a point in this. It is quite plausible that the owners might have increased their food intake while keeping their dogs’ usual feeding routines. We have now elaborated the discussion regarding both the food intake and the BCS-reduction in the manuscript. Moreover, the information regarding the BCS-assessments have been clarified in the method sections. All assessments were performed by the same veterinarian, with special expertise in the area. The reduction in BCS is also discussed in relation to results from a previous publication based on a slightly larger cohort from the same intervention, where it was shown that the dogs’ thigh circumference increased significantly while their bodyweight was stable, indicating a redistribution between body fat and muscle mass.

Response to Reviewer 2

We are deeply grateful for your kind words, and for your valuable feedback that really improved the manuscript.

Firstly, I consider the article to be methodologically constrained, primarily due to the small sample size. I acknowledge that conducting a study of this scope, involving multiple assessments that are highly dependent on participant compliance, requires a substantial level of commitment from dog owners throughout the entire project. Nevertheless, the limited sample size may introduce significant bias across the statistical analyses, potentially masking positive findings that may have been observed but did not reach statistical significance (p > 0.05). Although the challenges in maintaining a larger sample are understandable and clearly justified by the author, this limitation reduces the robustness of the results and highlights the need for further studies with larger samples to allow more definitive conclusions.

Yes, we completely agree and have tried to stress this more in the discussion and limitations sections.

Another point of concern relates to the eight-we

---

## [Decision Letter · Decision Letter 1]

25 Mar 2026

Effects of a joint outdoor exercise program for dog owners and dogs on physical activity, sedentary time and sleep-related behaviors

PONE-D-25-61363R1

Dear Dr. Smedberg,

We’re pleased to inform you that your manuscript has been judged scientifically suitable for publication and will be formally accepted for publication once it meets all outstanding technical requirements.

Kind regards,

Wolfgang Blenau

Academic Editor

PLOS One

Additional Editor Comments (optional):

Reviewers' comments:

Reviewer's Responses to Questions

**Comments to the Author**

Reviewer #1: All comments have been addressed

Reviewer #2: All comments have been addressed

2. Is the manuscript technically sound, and do the data support the conclusions?

Reviewer #1: Yes

Reviewer #2: Yes

3. Has the statistical analysis been performed appropriately and rigorously?

Reviewer #1: Yes

Reviewer #2: N/A

4. Have the authors made all data underlying the findings in their manuscript fully available?

Reviewer #1: Yes

Reviewer #2: Yes

5. Is the manuscript presented in an intelligible fashion and written in standard English?

Reviewer #1: Yes

Reviewer #2: Yes

Reviewer #1: (No Response)

Reviewer #2: (No Response)

.

Reviewer #1: No

Reviewer #2: **Yes:**Ricardo de Souza BuzoRicardo de Souza BuzoRicardo de Souza BuzoRicardo de Souza Buzo

---

## [Editor Report · Acceptance letter]

PONE-D-25-61363R1

PLOS One

Dear Dr. Smedberg,

I'm pleased to inform you that your manuscript has been deemed suitable for publication in PLOS One. Congratulations! Your manuscript is now being handed over to our production team.

Kind regards,

on behalf of

Dr. Wolfgang Blenau

Academic Editor

PLOS One